# Lipedema and the Potential Role of Estrogen in Excessive Adipose Tissue Accumulation

**DOI:** 10.3390/ijms222111720

**Published:** 2021-10-29

**Authors:** Kaleigh Katzer, Jessica L. Hill, Kara B. McIver, Michelle T. Foster

**Affiliations:** Department of Food Science and Human Nutrition, Colorado State University, 1571 Campus Delivery, 500 West Lake Street, Fort Collins, CO 80523, USA; kkatzer@rams.colostate.edu (K.K.); Jessica.Lynn.Hill@colostate.edu (J.L.H.); Kara.McIver@colostate.edu (K.B.M.)

**Keywords:** lipedema, adipose tissue, estrogen receptor, lipoprotein lipase, GLUT4, peroxisome proliferator-activated receptor gamma, adrenergic receptors, vascular endothelial growth factors, fat dysregulation, lower body subcutaneous adipose tissue

## Abstract

Lipedema is a painful fat disorder that affects ~11% of the female population. It is characterized by bilateral, disproportionate accumulation of subcutaneous adipose tissue predominantly in the lower body. The onset of lipedema pathophysiology is thought to occur during periods of hormonal fluctuation, such as puberty, pregnancy, or menopause. Although the identification and characterization of lipedema have improved, the underlying disease etiology remains to be elucidated. Estrogen, a key regulator of adipocyte lipid and glucose metabolism, and female-associated body fat distribution are postulated to play a contributory role in the pathophysiology of lipedema. Dysregulation of adipose tissue accumulation via estrogen signaling likely occurs by two mechanisms: (1). altered adipocyte estrogen receptor distribution (ERα/ERß ratio) and subsequent metabolic signaling and/or (2). increased release of adipocyte-produced steroidogenic enzymes leading to increased paracrine estrogen release. These alterations could result in increased activation of peroxisome proliferator-activated receptor γ (PPARγ), free fatty acid entry into adipocytes, glucose uptake, and angiogenesis while decreasing lipolysis, mitochondriogenesis, and mitochondrial function. Together, these metabolic alterations would lead to increased adipogenesis and adipocyte lipid deposition, resulting in increased adipose depot mass. This review summarizes research characterizing estrogen-mediated adipose tissue metabolism and its possible relation to excessive adipose tissue accumulation associated with lipedema.

## 1. Introduction

Lipedema is a fat disorder characterized by the bilateral and disproportionate accumulation of fat primarily within the lower body [1]. Lipedema is often misdiagnosed and misunderstood; however, improved characterization now distinguishes it from other commonly mistaken conditions such as lymphedema (swelling of the lower limbs) and, in most severe cases, general obesity. Distinctions are predominantly based on three factors, location of fat accumulation, timing of disease presentation, and the population affected [1]. For example, people with general obesity may experience fat accumulation at any point in life and deposition can occur throughout the body, while individuals with lipedema experience fat accumulation around times of hormonal shift and show distinct characteristics in the distribution of fat deposition [1]. Although lymphatic damage can occur with lipedema, the disease is not synonymous with lymphedema, a condition in which the lymphatic system is impaired resulting in abnormal swelling [1,2]. Lipedema occurs predominantly in women and estimates predict that it affects 11–19% of the female population [3]. However, these estimates are also thought to understate the true prevalence of disease given its frequency of misdiagnosis and late diagnosis [3,4,5]. Lipedema often presents during puberty, but may also appear or become exacerbated during pregnancy, menopause, or with the use of hormonal contraceptives [1,6]. Each of these circumstances involve alterations in hormone concentrations, especially estrogen. In support of this, only a small number of cases have been reported in men characterized with lower testosterone and/or increased estrogen levels [3]. Reports of lipedema in patient family histories suggest that it is likely a heritable disease and a genetic basis for the condition is also being investigated [4,7]. Thus far, a study of lipedema patient pedigrees has suggested an X-linked dominant pattern of inheritance, or more likely, an autosomal dominant inheritance with sex limitation [7,8]. No specific genetic cause has been identified for lipedema, but a recent review has compiled a list of genes which are associated with lipedema and other fat disorders, making them ideal targets for future research [8].

There are multiple classification systems utilized to characterize lipedema fat distribution, with some more widely used than others. The International Classification of Functioning, Disability and Health (ICF) system, created by the World Health Organization, was an early characterization method for lipedema classification based on body composition and patient functionality [9]. Other diagnostic criteria applied clinical classifications such as thickness of subcutaneous tissue to establish a more definitive description of the progression and stages of lipedema [10]. Yet, the Type (I–V) and Stage (1–4) classification systems (Table 1 and Table 2, respectively) are most widely accepted and recognized by lipedema organizations [2,4]. ‘Type’ describes the distribution of symmetrical adipose tissue deposition, while ‘Stage’ indicates changes in shape, consistency, texture and level of swelling [2,4]. Examples of ‘Stage’ alterations include (1.) palpable fat nodules (lipomas) that develop and increase in size with lipedema advancement (Table 2) [1,2,4], (2.) changes in the appearance and feel of the skin [1,2,4] and (3.) decreased vascular integrity that results in bruising quickly and easily [1,2,4,11].

Depending on Type and Stage, lipedema may be further associated with restriction in movement and pathophysiological outcomes. Specifically, some patients with lipedema report impaired mobility, which is postulated to result from increases in subcutaneous adipose tissue mass and limb swelling [2,11]. Pain, sometimes spontaneous, and tenderness in limbs are also linked to limbs diagnosed with lipedema [2]. Although the exact mechanisms causing limb pain/discomfort are currently unknown, some link it with subcutaneous tissue inflammation, damage, fibrosis, and nerve fiber abnormalities [2,3,11,12]. Lower limb swelling and edema are also commonly associated with lipedema [2,3,11] and are postulated to be secondary to the increase in adipose tissue. Although the mechanisms for this association remain to be elucidated, some predict it is due to tissue hypoxia induced by adipose tissue compression of small blood vessels. Compression of vessels disrupts vascular permeability resulting in increased protein and fluid in the interstitial space [13]. The lymphatic vessels, which aid in fluid removal, also become compressed by adipose tissue in lipedema, further inhibiting fluid removal from the lower limbs [13].

Despite current research efforts, treatment options for lipedema remain limited. Research supports those patients with lipedema describe weight loss to be difficult. Indeed, adipose tissue from lipedema patients is described to be “resistant” as it pertains to lipolysis mechanisms [3,11]. For most lipedema patients liposuction is described to be the most effective treatment for the removal of excessive adipose tissue, increasing patient mobility and decreasing edema [3]. Alternatively, there are less harsh treatments for edema control that include limb elevation, massage, exercise, and compression [2,13]. Although the identification and characterization of lipedema have improved, the underlying disease etiology remains to be elucidated. Multiple factors likely play a role in the development of lipedema; however, it is proposed that estrogen may contribute [12,14]. Briefly, lipedema occurs predominantly in women and is noted to progress and/or be exacerbated during times of hormonal fluctuation such as puberty, pregnancy and menopause and even with the use of hormonal contraceptives [1,6]. Men with lipedema have lower testosterone and/or increased estrogen levels [3,14]. Last, studies support that estrogen promotes fat accumulation specifically in the lower body adipose depots, which includes hips thighs and buttocks [15,16], which is a common site of excessive adipose tissue accumulation in most lipedema types (Table 1). Indeed, whole-exome sequencing indicates lipedema is associated with variants in genes encoding sex hormones involved in subcutaneous adipose tissue deposition [17].

Estrogen is a steroid hormone with three endogenous forms: estrone (E1), estradiol (E2), and estriol (E3). Estrogen signaling is primarily accomplished through two nuclear hormone receptors which, upon binding estrogen, modulate gene expression by directly binding DNA or contributing to transcription factor complexes [18]. Estradiol, the main product of estrogen synthesis, is the most potent form while estrone and estriol are weaker ligands [19]. Regulation of gene expression is facilitated by two types of estrogen receptors, ERα and ERß, with ERα being the predominate form [18,20]. Estrogen signaling is involved in numerous physiological pathways throughout the body including, but not limited to, reproduction, behavior, cardiovascular function, metabolism and body fat distribution [21]. Although most research is focused on ERα, both receptors have been shown to influence adipose tissue metabolism within human subcutaneous and mouse intra-abdominal depots [20,22,23,24]. Thus far, human research supports that estrogen is associated with accumulation of subcutaneous adipose tissue (between the skin and muscle) in the gluteal and femoral regions [25]. Indeed, the loss of estrogen with menopause shifts adipose tissue accumulation away from the lower body towards central/abdominal deposition [26], a distribution commonly seen in males [26,27]. The mechanisms, however, regulating estrogen-induced adipose tissue metabolism and subsequent distribution among different adipose depot regions, remain exceedingly controversial. This review summarizes research characterizing estrogen-mediated adipose tissue metabolism and its possible relation to excessive adipose tissue accumulation associated with lipedema. Subsequently, the concluding summary offers a prediction of how estrogen receptor dysregulation may play a role in the excessive lower body adipose tissue accumulation characterized in lipedema.

## 2. Role of Estrogen Signaling through ERα

Estrogen regulates many aspects of lipid and glucose metabolism, yet the precise role of ERα in adipose tissue accumulation remains controversial. Early investigations in rodents suggested that ERα inhibits adipose tissue accumulation. The role of estrogen/ERα signaling was first demonstrated in ERα whole-body knockout mice (αERKO) [28], where both male and female mice were prone to increased adiposity. Other studies utilizing fat pad-specific injections (perigonadal) of adeno-associated ERα siRNA virus support these findings. Site-specific knockdown of ERα causes adipocyte hypertrophy and consequently increased depot mass [29]. Some human studies, however, oppose these findings. Specifically, studies in pre-menopausal women describe estrogen to promote body adipose tissue growth, lower body subcutaneous specifically [30,31,32,33]. We postulate that the discordance among observations over ERα adipose tissue function likely result from differences in species and adipose depot location studied within the body. Differential outcomes are also likely to occur in studies that utilize adipocyte cell lines. The sections below will summarize and discuss these controversial studies and associated factors pertinent to ERα-mediated adipocyte and adipose depot metabolism.

### 2.1. Lipoprotein Lipase

Lipoprotein lipase (LPL) catalyzes the rate-limiting step in the hydrolysis of circulating lipoproteins [34]. In adipose tissue, LPL is synthesized by adipocytes, but is inactive until it is translocated to the luminal side of endothelial cells lining blood vessels [35]. This enzyme permits free fatty acids to enter adipocytes for the production of triacylglycerol (TAG) molecules [34]. Estrogen signaling is demonstrated to regulate adipose depot LPL synthesis. Research, however, is controversial, with arguments supporting that estrogen decreases LPL function, while others demonstrate LPL function is not altered by estrogen treatment.

In-depth studies demonstrating a fundamental role of estrogen in adipose tissue lipid metabolism occur in rodents, cell lines and humans. Some of these studies propose that adipose tissue lipid accumulation is inhibited by estrogen treatment. For example, studies demonstrate that LPL production decreases in gonadal adipose tissue (periovarian—fat surrounding the ovaries, fallopian tubes and uterus) of female rats treated with estrogen [36]. This is further supported in ovariectomized (OVX) mouse model of menopause. Specifically, estrogen (E2) treatment prevents periovarian adipose accumulation in OVX mice by decreasing gene expression of Lpl and other targets [37]. Rodent model findings are also supported by in vitro studies using 3T3-L1 cells, where LPL production is inhibited by nuclear activity of ERα [38]. Furthermore, estrogen treatment of 3T3-L1 cells, genetically altered to continuously express Esr1, decreases Lpl transcription via suppression of the Lpl gene promoter [38]. Overall, these studies indicate that estrogen signaling in the gonadal depot of rodents and in vitro cells decreases fatty acid entry into adipocytes, hence decreases lipid accumulation.

Although the rodent and cell models support that estrogen inhibits LPL by several mechanisms, human studies remain controversial and contradictory. For example, some human studies report LPL mRNA expression in female subcutaneous adipose tissue does not change following estrogen treatment [39]. In agreement, another study found no difference in LPL activity in the subcutaneous adipose tissue of pre- and postmenopausal women [40]. Others, however, indicate that estrogen decreases LPL activity in gluteal fat [41]. In support of this, premenopausal women with gluteal region transdermal 17β-estradiol had significantly decreased LPL activity in adipose biopsies collected beneath the patches [42]. Yet, oral treatment with ethinyl estradiol in postmenopausal women increases gluteal adipose tissue LPL activity levels [43]. In agreement with this, studies demonstrate that basal lipolysis is 77% lower in gluteal adipose cells from postmenopausal compared with perimenopausal women [44]. Interindividual and study method differences likely drive the contradictory outcomes of estrogen regulation of adipose tissue LPL activity. Additionally, when comparing rodent study outcomes to those of human, it is important to note that fat depots are not all inherently the same, gonadal fat regulation is not analogous to gluteal.

### 2.2. Adipogenesis via Peroxisome Proliferator-Activated Receptor Gamma (PPARγ)

Peroxisome proliferator-activated receptor gamma (PPARγ) is one isoform member of the peroxisome proliferator-activated receptor family. These proteins function within the nucleus to modulate gene expression through direct binding of DNA [45]. The PPARγ isoform is well known as a master regulator of glucose and lipid metabolism within many tissues, but especially skeletal muscle and adipose tissue [46,47]. Signaling through PPARγ stimulates TAG synthesis and storage within adipocytes, contributing to increased adipocyte size [45]. It also activates adipogenesis, the differentiation of pre-adipocytes into mature adipocytes [45,47]. Ultimately, PPARγ promotes the expansion of fat mass by facilitating the production of mature adipocytes and contributing to their lipid deposition. Research demonstrates that ERα contributes to alterations in adipose tissue mass via PPARγ; however, results are controversial as to whether ERα inhibits or activates PPARγ. Respective studies are discussed.

ERα action has been demonstrated to inhibit Pparg expression and activity in rodent models. A study in ovariectomized female rats demonstrated that that ERα agonist treatment reduced periovarian adipose tissue Pparg expression [36]. Similarly, ERα agonist treatment reduces PPARG expression in human cancer cell lines [48]. Specifically, ERα-repressed/deficient cells exhibited greater PPARG expression than untreated cells, reinforcing evidence for the inhibitory effect of ERα on PPARG expression [48]. Together, these studies demonstrate that ERα activation inhibits PPARγ activity, which would consequently decrease adipogenesis and storage of TAG’s within mature adipocytes and reduce overall adipose tissue mass. In opposition, others demonstrate that estrogen upregulates PPARG expression. Human adipose-derived stromal cells matured in an adipogenic differentiation medium supplemented with estradiol are reported to have greater ERS1 and PPARG expression than those without estradiol treatment [49]. A later study using 3T3-L1 mouse adipocytes supported these results by demonstrating an increase in PPARγ protein concentration with chronic exposure to estradiol, although it was not identified which estrogen receptor was involved [50]. Overall, these studies suggest that ERα activation stimulates PPARG expression, which in adipose tissue would lead to an increase in adipogenesis and lipid deposition within adipocytes, increasing overall adipose tissue mass. Although the literature presents conflicting reports on the direction of ERα control over PPARγ activity, it is clear that a regulatory relationship exists between the two proteins and it should remain an interest of lipedema research.

### 2.3. Lipolysis/Lipogenesis via Adrenergic Receptors (ARs)

Catecholamines such as epinephrine and norepinephrine act on tissues through adrenergic receptors (ARs), which are split into α and ß classes, and have been shown to regulate lipolysis through the inhibition or activation of cAMP and the phosphorylation of lipolytic enzymes such as hormone-sensitive lipase [51,52]. α and ß ARs have reciprocal effects on lipolysis: ßAR’s stimulate lipolysis, while αAR’s are anti-lipolytic and positively correlate with an increase in adipose tissue mass [51,53,54]. Because men and women have markedly different fat accumulation patterns, it should be noted that the relative expression of αAR and ßAR varies by adipose depot [53,54]. Men tend to store excess fat within visceral adipose depots while women store more fat within subcutaneous adipose depots, especially the gluteal-femoral region [15,16,54].

Gluteal-femoral subcutaneous adipose tissue, typically larger in women, demonstrates a greater αAR response and lower ßAR response than abdominal subcutaneous adipose tissue or visceral adipose tissue [53,54]. Conversely, visceral adipose tissue, which is typically larger in men, exhibits a greater ßAR response and lower αAR response than subcutaneous adipose tissue [53,54]. In support of these observations, pre-menopausal women are shown to have a higher density of adipocyte αAR than men [53]. Taken together, differences in AR between adipose depots likely contributes to the variant lipolysis rates and distribution of adipose tissue between women and men. Interestingly, estrogen has been shown to regulate ADR expression through ERα activation [51,53,54]. In vivo and in vitro studies using human abdominal subcutaneous adipose tissue have demonstrated that ERα activation by estrogen upregulates the expression of ADRA2A [53]. More so, the increase in ADRA2A expression within subcutaneous adipocytes led to an attenuation of lipolysis [53]. This same study also revealed that stimulation with estrogen had no effect on ADRA2A expression within visceral adipose depots. These findings may help explain the gender differences in adipose tissue distribution and how estrogen drives gluteal/femoral storage at the expense of visceral accumulation in women. Additionally, inhibition of lipolysis by these receptors may be a key factor in why lipedema patients are unable to lose weight despite lowering caloric intake. Indeed, higher αAR/ßAR ratios and ßAR polymorphisms are correlated with general obesity [55]. Further research is needed to clarify the interaction between ERα with βARs.

### 2.4. Glucose Transporter 4 (GLUT 4) and Glucose Translocation

The glucose transporter 4 (GLUT4), found predominantly on skeletal muscle cells and adipocytes, plays a central role in glucose metabolism [56]. Insulin released in response to elevated blood glucose stimulates the translocation of GLUT4 transporters from intracellular vesicles to the cell membrane [56]. Glucose is then passively transported through GLUT4 and into the cell for utilization or storage. Estrogen has been shown to modulate GLUT4 through both ERα, a positive regulator, and ERß, a negative regulator. The role of ERα in glucose transport is supported by ERα −/− mouse models, which demonstrate a reduced expression of Slc2a4 in both gonadal white adipose tissue and gastrocnemius muscle [22,57]. Upregulation of GLUT4 retains insulin tolerance in these ERα-deficient tissues [20,22]. Similarly, two studies demonstrate in differentiated 3T3-L1 adipocytes that estradiol increases AKT phosphorylation [58], Slc2a4 expression [58,59], insulin-stimulated GLUT4 translocation and glucose uptake [58]. Overall, ERα increases GLUT4 and may contribute to exacerbated fat tissue accumulation within lipedema patients through the increased movement and storage of glucose within adipocytes.

### 2.5. Angiogenesis via Vascular Endothelial Growth Factors (VEGF)

Vascular endothelial growth factors (VEGFs) are a family of secreted proteins which act through receptor kinases to initiate angiogenesis, the branching of new capillaries from an existing vascular bed [60]. In fat angiogenesis is necessary to maintain adipose tissue health while also promoting growth and expansion [61]. Initial studies investigating the relation between estrogen and VEGF were conducted in breast tissue within the context of cancer research. These investigations reported an upregulation of VEGFA expression with estrogen treatment, indicating the importance of estrogen in VEGF regulation [62,63]. Specifically, in human breast cancer cell lines MCF-7 and MDA-MB-231, the addition of estrogen was shown to positively correlate with an increase in VEGFA expression [62]. Additionally, healthy human breast tissue treated with estradiol led to an increase in extracellular VEGF levels [63]. A 2017 study utilizing 3T3-L1 cells and ERα knockout mice confirmed that estrogen was a positive regulator of Vegfa expression via ERα [61]. Specifically, in 3T3 cells, ERα agonists induce Vegfa expression and inhibition of ERα led to VEGF reduction. In support of this, Vegfa expression in inguinal and gonadal adipose tissues in ERα knockout mice is lower than wild-type mice [61]. Maintenance of adequate blood supply by ERα signaling sustains adipose tissue health. ERα is a positively associated with VEGF protein release, hence estrogen signaling via angiogenesis can contribute to adipose tissue growth and expansion.

## 3. Role of Estrogen Signaling through ERß

To date, estrogen signaling through ERß has not been investigated as thoroughly as ERα. Like ERα, ERß regulates glucose and lipid metabolism by modulating gene expression. Specifically, regarding adipose tissue, ERβ and its selective ligands are demonstrated to differentiate mesenchymal stem cells toward a brown adipocyte phenotype rather than white [64]. This indicates that a function of ERβ activation in adipose tissue is to increase thermogenic capacity where the oxidation of fat is dissipated as heat rather than energy as ATP. Consistent with this, adipocyte ERβ activation also increases mitochondria biogenesis and mitochondrial function in mature brown adipocytes [64]. These outcomes indicate that ERβ in adipose tissue enhances fat/lipid burning process. It is generally accepted that ERß signaling acts in opposition to ERα [20,22]. However, further research is needed to elucidate the role of ERβ in lipid homeostasis.

ER subtypes, α and ß, do not demonstrate equal expression within adipose tissue. For example, characterization of abdominal and omental adipose tissues from healthy individuals found a greater expression of ERα than ERß [24]. In support, a recent study noted a higher ERα/ERß ratio in abdominal subcutaneous adipose tissue than in gluteal fat from overweight pre-menopausal women [65]. Others also demonstrate a positive correlation between the ratio of ERα/ERß in adipose depots and obesity/enhanced adiposity [66]. Altogether these results suggest that the ERα/ERß ratio may be altered in obesity or other disease states. It is worth noting that there may also be variable ESR2 expression between men and women. One group reported greater overall ESR2 expression in intraabdominal (omental) and subcutaneous (inguinal) depots of women but no difference in ESR1 expression, giving females a lower ERα/ERß ratio than men [24]. Thus, it is plausible that variability in the ratio of ERα/ERß may also contribute to differences in fat distribution between men and women.

### 3.1. Inhibitor of ERα

Early studies on the interaction between ERα and ERß within human breast cancer cells demonstrated that ERß downregulates the expression of genes controlled by ERα; however, the mechanisms remained to be elucidated [67]. Investigations extend that ERß indirectly controls ERα activity. Specifically, ERß was found to modify ESR1 expression through targeted degradation by proteasomes, therefore inhibiting ERα-mediated gene regulation [68]. Additionally, others demonstrate in human breast cancer cells that ERß inhibits ESR1 expression through binding to the Sp-1 transcription factor and recruiting a co-repressor to the ERα gene promoter, effectively halting transcription [69]. However, further examination of this regulation should be conducted in the context of adipose tissue to better understand its potential role in lipedema pathophysiology.

### 3.2. PPARγ Activity and Adipogenesis

As previously discussed, PPARγ is a master regulator of lipid metabolism. Signaling through this receptor stimulates TAG storage within adipocytes and the activation of adipogenesis, ultimately supporting the expansion of adipose tissue. Regulation of PPARγ by ERß is less controversial than ERα and most reports support a reciprocal relationship. ERß knock-out mice were shown to have higher PPARγ activity in the gonadal depot, resulting in greater fat mass [20,46]. The same group studied 3T3-L1 cells in vitro, demonstrating that ERß inhibited ligand-mediated PPARγ activation [46]. Conversely, overexpression of ESR2 in human thyroid cancer cells resulted in reduced PPARγ activity [48]. These results are supported by a study in ovariectomized rats which found decreased gonadal fat Pparg expression after the addition of an ERß agonist [36]. Reduced ERß could result in excess fat accumulation seen in lipedema as greater PPARγ activity resulting in increased adipogenesis and TAG storage leading to exacerbated fat tissue accumulation.

## 4. Coregulators of the Estrogen Receptor

Regulation of gene transcription by nuclear receptors, including the estrogen receptor, involves not only ligand binding but also the recruitment of coregulators. They are characterized to critically contribute to the proper function of ER signaling and activity in physiology, development, and reproduction. ER coregulators are also linked to cancer and diseases. Hence, it is possible that coregulators play a role in the excessive adipose tissue accumulation associated with lipedema. Coregulators are demonstrated to be crucial for ER transcriptional activity, as molecules that contribute to the formation of large protein complexes to modulate appropriate activity on target gene chromatin [70]. Coregulators are also capable of regulating target gene transcription via acetylation, methylation, and ubiquitination, much like an enzyme would [70]. Steroid hormone receptors ERα and ERβ function may be positively or negatively influenced by coregulators. Those that increase transcriptional activity of steroid hormone receptors are coactivators and corepressors decrease it. Steroid receptor coactivator SRC is among the earliest discovered coregulator of Erα [71]. Since then, a large number of ERα coregulators have been identified and to a lesser extend coregulators of ERβ [72,73] Estrogen receptor coregulators and pioneer factors: the orchestrators of mammary gland cell fate and d. Taken together, ER gene transcription response to the estrogen ligand is critically regulated by the coregulator present [73]. The distinct differential outcomes of ERα and ERβ signaling may likely be due to differing response to and/or differential utilization of coactivators and corepressors.

## 5. Estrogen Synthesis Pathway

Downstream signaling of estrogen within adipose tissue via the estrogen receptor likely contributes to the pathophysiology of lipedema; however, this may not be an independent contributor. Dysregulation of estrogen synthesis may also contribute to the excessive accumulation of lower body adipose tissue. Estrogen synthesis predominantly occurs in female reproductive structures such as the ovaries, corpus luteum, and the placenta [19]. Although estradiol production is generally thought of as an endocrine product of the ovary, numerous other tissues have the capacity to synthesize estrogens from androgens and use estrogen in a paracrine or intracrine fashion. Indeed, adipose tissue (AT) is an organ characterized to produce estrogens and can contribute significantly to the circulating pool of estrogens [74,75,76,77]. There are three primary forms of estrogen: estrone (E1), estradiol (E2), and estriol (E3). Estradiol, the main product of estrogen synthesis, is the most potent form while estrone and estriol are weaker ligands [19]. Figure 1 outlines the general estrogen synthesis pathway including conversions between estrogen forms. All steroid synthesis pathways begin with cholesterol, the common precursor molecule, and are altered through successive enzymatic conversions. The first and rate-limiting enzymatic step of the steroidogenic process is the conversion of cholesterol into pregnenolone, which is subsequently converted to progesterone. P450C17 is needed to initiate androgen and estrogen synthesis [74]. P450c17 can mediate testosterone biosynthesis via the conversion of pregnenolone to dehydroepiandrosterone or via conversion of progesterone to androstenedione [78]. The final step in physiological synthesis of estradiol is aromatization of precursor testosterone by a CYP 19 gene product, cytochrome P450 estrogen aromatase. estradiol activity can be regulated by SULTE1E via sulfoconjugation leading to inactivation [19] or estradiol can be converted to the other estrogen forms. Estradiol and estrone can be interconverted with the action of 17β-hydroxysteroid dehydrogenases (17β-HSDs), whereas Estradiol can be hydroxylated into estriol via the enzyme Cytochrome P450 3A4 (CYP3A4). Overall, a dysregulation in adipose tissue sex steroid hormone production and paracrine signaling within the depot may contribute to the excessive accumulation of lower body adipose tissue characterized in lipedema.

## 6. Discussion

Lipedema is a painful fat disorder which predominantly affects women and develops during times of hormonal fluctuation including, but not limited to, puberty, pregnancy, and menopause [1,6]. As awareness of lipedema grows, there has been progress in the characterization of disease progression from a phenotypic standpoint and a collection of classification systems have been developed to describe disease progression [2]. The inciting and exacerbating factors of lipedema pathophysiology remain to be elucidated despite an increasing awareness for the disease. Here, we predict that because lipedema development occurs predominantly in woman during times of hormonal modifications, estrogen signaling likely plays a role.

The bilateral and disproportionate accumulation of fat primarily within the lower body characterized in lipedema [1] is likely, in part, associated with adipose estrogen dysregulation. Indeed, estrogen signaling in adipose tissue is demonstrated to alter lipid accumulation, fatty acid uptake, lipogenesis and subsequently adipocyte/adipose depot size. This estrogen signaling among different adipose tissue depots regulates and is a key feature of the sexual dimorphisms observed in body fat accumulation and adipose tissue expandability between males and females. As such, many pre-menopausal females accumulate fat in the subcutaneous adipose depots (predominantly lower body). We postulate that estrogen receptor signaling and/or adipocyte production of estrogen represents an early critical node in the development of adipose tissue dysfunction associated with lipedema.

Although studies investigating the role of estrogen in adipose tissue metabolism remain controversial, we postulate that ERα and ERß may play a role in the dysregulated adipose tissue characterized in lipedema. This possible estrogen-mediated dysregulation proposed to be associated with excessive lower body subcutaneous fat accumulation characterized in lipedema may occur by two mechanisms. First, adipocytes in this region from individuals with lipedema may be characterized with a higher ERα/ERß ratio compared to those without lipedema. This could result in (1.) reduced ERß-induced suppression of ERα-mediated expression of genes, (2.) reduced ERß inhibition and an increase in ERα activation of PPARγ, (3.) increased free fatty acid entry into adipocytes for the production of triacylglycerol via increased LPL activity, (4.) decreased lipolysis due to ERα-induced upregulation of the anti-lipolytic αAR, (5.) increased ERα-induced glucose uptake via enhanced insulin stimulated GLUT4 translocation, (6.) increased angiogenesis via ERα-induced increases in VEGF, and (7.) reduced mitochondriogenesis and mitochondrial function. Second, in lipedema there may also be an enhanced production of steroidogenic enzymes produced by adipocytes contained within the lower body subcutaneous adipose depot. Paracrine signaling would occur among adipocytes within this region activating ERα. Taken together, these outcomes would result in an increase in adipogenesis and lipid deposition within adipocytes, increasing overall adipose tissue mass.

Without an understanding of the mechanisms through which lipedema advances, researchers are limited in their ability to develop potential medicinal treatments or cures. Therefore, future research must focus on uncovering the etiology of lipedema to expand treatment prospects and improve the quality of life for individuals with this disease. Adipose tissue regulation is likely a great area of investigation in lipedema.

## Figures and Tables

**Figure 1 ijms-22-11720-f001:**
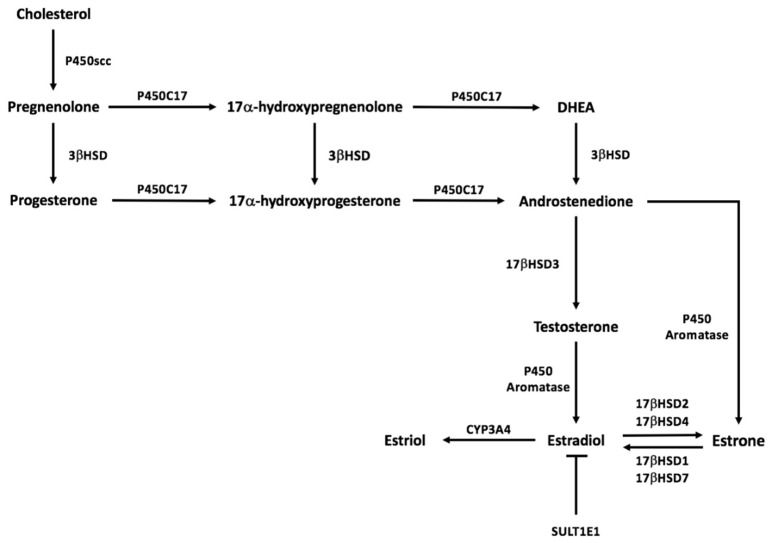
Sex steroid synthesis pathway.

**Table 1 ijms-22-11720-t001:** Characterization of lipedema type.

Type	Fat Location
Type 1	Pelvis, buttocks, hips
Type 2	Buttocks to knees, with formations of folds of fat around the inner side of the knee
Type 3	Buttocks to ankles
Type 4 a–c	(a) Upper arm (b) Lower arm (c) Whole arm
Type 5	Knees to ankles

The type of disease is based on the location of adipose tissue accumulation. Table is adapted from the Lipedema Foundation [2].

**Table 2 ijms-22-11720-t002:** Characterization of lipedema by stage.

Stage	Disease Progression
Stage 1	Normal skin surface with enlarged subcutaneous tissue; fat tissue is soft with noticeable small nodules
Stage 2	Uneven skin with enlarged subcutaneous tissue; larger fat nodules present
Stage 3	Large extrusions of tissue causing deformations, especially on the thighs and around the knees; fat nodules of varying sizes are palpable
Stage 4	Development of lipolymphedema with large overhangs of tissue

The stage of disease is based on the progression of fat accumulation and changes to the skin and lymphatic system. Lipolymphedema is a condition in which the increased fat accumulation and strain in the lymphatic system results in lymphedema in addition to lipedema. Table is adapted from the Lipedema Foundation [1,2].

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
