# Peer review of "Lipedema and the Potential Role of Estrogen in Excessive Adipose Tissue Accumulation"

_ijms, 2021, doi:10.3390/ijms222111720_

Round 1

Reviewer 1 Report

The manuscript by Katzer et al reviews the role of estrogen in excessive adipose accumulation in lipedema. The topic is interesting however there are similar recent reviews published (10.5772/intechopen.96402) by other authors. The strengths of this review over the previous published article is not mentioned or seen in the manuscript.

Author Response

Reviewer 1 has raised a significant concern as to the similarity of the submitted work to a work found at https://www.intechopen.com/online-first/75320. Your submission is more expansive than that work, but due to the similarity and timing, please emphasize in your abstract what makes yours different - likely you more expansive discussion of estrogen action on adipose function. Increasing discussion of estrogen actions on adipocyte health, function, and inflammation may be of use or including references to any sexual dimorphism in adipose fibrosis or vascular health may be warranted even if estrogen was not specifically measured. 

Comment Reviewer 1: The authors of the work, https://www.intechopen.com/online-first/75320, are fellow collaborators from the Lipedema foundation. For the foundation we are highly encouraged to bring awareness to the disease. As such, I was invited by Rachelle Crescenzi and Joseph Rutkowski, also grantees from the Lipedema foundation, to write this review for this special addition in this journal, to promote awareness for the lipedema foundation. Although the overall topic of estrogen may be the same, the way it is discussed is completely different than the publication by Sarah Al-Ghadban, we take a much deeper dive into the studies showing that data about adipose tissue accumulation or lipolysis is controversial and not straight forward. The paper by Al-Ghadban had broad short generalization with minimal details of these topics and focused across many topics including leptin, insulin, inflammation, fibrosis, hormone therapy and associated disease. These extra topics are not the focus of our paper we submitted.  

Given that inflammation was a focus of the Al-Ghadban publication I think it is best to not included that in our publication or other associated factors like fibrosis. Our focus is only on the adipose tissue accumulation aspects and the studies that demonstrate that estrogen may play a role in lipogenesis or lipolysis. We are focusing only on the drivers of the excessive adipose tissue accumulation associated with lipedema. Our introduction takes the same approach of characterizing the disease because that description is standard for our foundation. Past that point we take a much deeper dive into only the lipid regulation aspect, this publication is much more specific and expansive.

Our paper also includes an opposing view, they say estrogen should be a therapy, whereas we postulate it is the cause of the disease and exacerbates accumulation, we actually give a postulate based off of the more detail look into studies (this is detailed in our abstract), this is provided in the discussion and is our original thoughts as we proposed to the lipedema foundation.

Our paper is a 10 page expansion of what they summarize in 2 double spaced paragraphs under the heading estrogen and adipogenesis (2.1).

This is their abstract:

Lipedema is an underdiagnosed painful adipose tissue disorder that occurs almost exclusively in women, with onset manifesting at puberty or at times of hormonal change. Unlike many fat disorders, diet and exercise have little to no impact on the prevention or progression of this disease. Estrogens control the distribution of body fat and food intake, regulate leptin expression, increase insulin sensitivity, and reduce inflammation through signaling pathways mediated by its receptors, estrogen receptor alpha (ERα) and ERβ. This review will focus on understanding the role of estrogen in the pathogenesis of the disease and envisage potential hormonal therapy for lipedema patients.

Whereas this is ours (we are not discussing hormone therapy, if anything this is a detail argument against it):

Lipedema is a painful fat disorder that affects ~11% of the female population. It is characterized by bilateral, disproportionate accumulation of subcutaneous adipose tissue predominantly in the lower body. The onset of lipedema pathophysiology is thought to occur during periods of hormonal fluctuation, such as puberty, pregnancy, or menopause. Although the identification and characterization of lipedema has improved, underlying disease etiology remains to be elucidated. Estrogen, a key regulator of adipocyte lipid and glucose metabolism, and female associated body fat distribution are postulated to play a contributory role in the pathophysiology of lipedema. Dysregulation of adipose tissue accumulation via estrogen signaling likely occurs by two mechanisms: 1.) altered adipocyte estrogen receptor distribution (ERα/ERß ratio) and subsequent metabolic signaling and/or 2.) increased release of adipocyte produced steroidogenic enzymes leading to increased paracrine estrogen release. These alterations could result in increased activation of peroxisome proliferator-activated receptor γ (PPARγ), free fatty acid entry into adipocytes, glucose uptake, and angiogenesis while decreasing lipolysis, mitochondriogenesis, and mitochondrial function. Together these metabolic alterations would lead to increased adipogenesis and adipocyte lipid deposition, resulting in increased adipose depot mass. This review summarizes research characterizing estrogen mediated adipose tissue metabolism and its possible relation to excessive adipose tissue accumulation associated with lipedema.

It is my firm belief that our publication is indeed very different than our collaborations reviewer 1 is referring to.

Reviewer 2 Report

Dear Authors, thanks for this review interesting and easy to read about lipodema and estradiol.

I have two minors comments about, I think, two small but important forgotten points in your review.

First, lipedema is characterized by adipocyte hyperplasia and hypertrophy. The pathways described in our review are essentially related to hypertrophy and adipocyte function. Some sentences should be introduced about the role of ERa and ERb in adipogenesis, following for example their role in PPARg expression which coordinates in part adipogenesis.

Second, a paragraph about ER coactivator and corepressor should be added, as they are highly important in ER signaling and adipocyte biology.

Author Response

There are overviews related to adipogenesis and then we break it down into the discussion of factors that play a role such as PPAR, LPL, GLUT4, AR. First for Era:

  1. Role of Estrogen Signaling Through ERα

Estrogen regulates many aspects of lipid and glucose metabolism, yet the precise role of ERα in adipose tissue accumulation remains controversial. Early investigations in rodents suggested that ERα inhibits adipose tissue accumulation. The role of estrogen/ERα signaling was first demonstrated in ERα whole body knockout mice (αERKO) [1], where both male and female mice were prone to increased adiposity. Other studies utilizing fat pad-specific injections (perigonadal) of adeno-associated ERα siRNA virus support these findings. Site-specific knockdown of ERα causes adipocyte hypertrophy and consequently increased depot mass [2]. Some human studies, however, oppose these findings. Specifically, studies in pre-menopausal women describe estrogen to promote body adipose tissue growth, lower body subcutaneous specifically [3-6]. We postulate that the discordance among observations over ERα adipose tissue function likely result from differences in species and adipose depot location studied within the body. Differential outcomes are also likely to occur in studies that utilize adipocyte cell lines. The sections below will summarize and discuss these controversial studies and associated factors pertinent to ERα mediated adipocyte and adipose depot metabolism.

Then ERB:

  1. Role of Estrogen Signaling Through ERß

To date, estrogen signaling through ERß has not been investigated as thoroughly as ERα. Like ERα, ERß regulates glucose and lipid metabolism by modulating gene expression. Specifically, regarding adipose tissue, ERβ and its selective ligands are demonstrated to differentiate mesenchymal stem cells toward a brown adipocyte phenotype rather than white [7]. This indicates that a function of ERβ activation in adipose tissue is to increase thermogenic capacity where the oxidation of fat is dissipated as heat rather than energy as ATP. Consistent with this, adipocyte ERβ activation also increases mitochondriogenesis and mitochondrial function in mature brown adipocytes [7]. These outcomes indicate that ERβ in adipose tissue enhances fat/lipid burning process. It is generally accepted that ERß signaling acts in opposition to ERα [8,9]. However, further research is needed to elucidate the role of ERβ in lipid homeostasis.

Second, a paragraph about ER coactivator and corepressor should be added, as they are highly important in ER signaling and adipocyte biology.

This information has been added to the paper as written below:

Regulation of gene transcription by nuclear receptors, including the estrogen receptor, involves not only ligand binding but also the recruitment of coregulators. They are characterized to critically contribute to the proper function of ER signaling and activity in physiology, development, and reproduction. ER coregulators are also linked to cancer and diseases. Hence, it is possible that coregulators play a role in the excessive adipose tissue accumulation associated with lipedema. Coregulators are demonstrated to be crucial for ER transcriptional activity, as molecules that contribute to the formation of large protein complexes to modulate appropriate activity on target gene chromatin [10]. Coregulators are also capable of regulating target gene transcription via acetylation, methylation, and ubiquitination, much like an enzyme would [10]. Steroid hormone receptors ERα and ERβ function may be positively or negatively influenced by coregulators. Those that increase transcriptional activity of steroid hormone receptors are coactivators and corepressors decrease it. Steroid receptor coactivator SRC is among the earliest discovered coregulator of Erα [11]. Since then, a large number of ERα coregulators have been identified and to a lesser extend coregulators of ERβ [12,13]  Estrogen receptor coregulators and pioneer factors: the orchestrators of mammary gland cell fate and d. Taken together, ER gene transcription response to the estrogen ligand is critically regulated by the coregulator present [13]. The distinct differential outcomes of ERα and ERβ signaling may likely be due to differing response to and/or differential utilization of coactivators and corepressors.